# Educator-Informed Development of a Mental Health Literacy Course for School Staff: Classroom Well-Being Information and Strategies for Educators (Classroom WISE)

**DOI:** 10.3390/ijerph20010035

**Published:** 2022-12-20

**Authors:** Jaime C. Semchuk, Shannon L. McCullough, Nancy A. Lever, Heather J. Gotham, Jessica E. Gonzalez, Sharon A. Hoover

**Affiliations:** 1BC Children’s Hospital, Vancouver, BC V6H 3N1l, Canada; 2WestEd, 730 Harrison Street, San Francisco, CA 94107, USA; 3National Center for School Mental Health, University of Maryland School of Medicine, 737 West Lombard Street, 4th Floor, Baltimore, MD 21201, USA; 4Mental Health Technology Transfer Center, Network Coordinating Office, Stanford University School of Medicine, 1520 Page Mill Road, Palo Alto, CA 94304, USA

**Keywords:** mental health literacy, educator mental health training, professional development

## Abstract

Educators play a critical role in promoting mental health and well-being with their students. Educators also recognize that they lack knowledge and relevant learning opportunities that would allow them to feel competent in supporting student mental health. As such, educators require resources and training to allow them to develop skills in this area. The Mental Health Technology Transfer Center (MHTTC) Network partnered with the National Center for School Mental Health at the University of Maryland School of Medicine to develop Classroom Well-Being and Information for Educators (WISE), a free, three-part mental health literacy training package for educators and school staff that includes an online course, video library, and resource collection. The Classroom WISE curriculum focuses on promoting positive mental health in the classroom, as well as strategies for recognizing and responding to students experiencing mental health related distress. This paper describes the curriculum development process, including results of focus groups and key informant interviews with educators and school mental health experts. Adoption of Classroom WISE can help educators support student mental health and assist in ameliorating the youth mental health crisis.

## 1. Introduction

Schools are increasingly recognized as a critical venue for promoting student mental health and well-being and addressing mental health challenges via school-based mental health services [1]. Mental health is often defined and understood in terms of the presence or absence of mental illness. However, increasingly, definitions of mental health incorporate discussion of both the presence of positive indicators of wellbeing (e.g., life satisfaction, self-acceptance, positive emotions), as well as the absence of disorder [2,3,4]. With youth mental illness and suicide rates increasing over the last decade, and exacerbated by COVID-19, states and local communities have embraced school mental health as a mechanism for addressing what many are declaring as a “youth mental health crisis” [5,6]. Federal agencies and national technical assistance centers have encouraged state and local education authorities to utilize COVID-19 relief funds, in part, to support student mental health and well-being [7,8]. School staff education about mental health, including how to promote student well-being and identify and support students in distress, is considered foundational to implementing school mental health efforts [1,9]. In an effort to offer standardized, evidence-informed mental health training for educators in the United States, the federal Substance Abuse and Mental Health Services Administration (SAMHSA) charged the Mental Health Technology Transfer Center (MHTTC) Network with developing resources for educators. The MHTTC Network partnered with the National Center for School Mental Health (NCSMH) at the University of Maryland School of Medicine to develop an online mental health training package for school staff. This paper describes the development of the package, Classroom Well-Being and Information for Educators (WISE), including background on the need for the course and how results of focus groups and key informant interviews were used in its development. 

### 1.1. Educator Roles in Comprehensive School Mental Health Systems

A population-based, public health, or multi-tiered system of support (MTSS) approach to school mental health offers a theoretical framework and rationale for providing educator-focused mental health literacy professional development as part of the foundation for supporting student mental health in schools. Often referred to as Comprehensive School Mental Health Systems, the population-based, MTSS model is consistent with ecological and community mental health frameworks [1,10]. When compared to a traditional focus on the mental health of individual clients adhered to by clinical psychology, the public health model can be understood to address clients across the broader population [10,11]. As such, targeting prevention of mental health problems and promotion of mental health and wellbeing at the universal level is balanced with providing targeted services to individuals at-risk for mental health problems, as well as serving those clients with significant mental health needs [11,12]. Furthermore, a population-based, MTSS approach focuses on bolstering positive behaviors and strengthening connections across various community sectors to create integrated systems of care that serve the broad population [10,11].

Within a population-based, MTSS mental health framework, at the universal level (i.e., tier one), mental health promotion strategies are focused on the general population. Within the school setting, universal strategies and interventions may include school-wide positive behavior support initiatives, social emotional learning, and other mental health promotion programs and curricula that can be delivered by educators and school teams that serve the entire student population [13]. At the targeted level (i.e., tier two), individuals at risk for mental health problems are identified and supported through targeted prevention interventions and risk or harm reduction strategies. In the school context, educators help to identify students at-risk for mental health problems in need of further intervention and provide classroom adaptations to support students and promote mental wellbeing, such as trauma-informed behavioral supports (activities that recognize the possible impact of trauma and are sensitive to avoid re-traumatizing the person) [13]. One example of a trauma-informed behavioral support is the implementation of consistent and predictable classroom routines. At the indicated level (i.e., tier three), individuals with mild to complex mental health problems are treated with therapeutic interventions at varying levels of intensity as indicated by the individual’s specific mental health needs. Within schools, educators support indicated services by participating in referrals and support interventions, such as the implementation of crisis response protocols [13]. Crisis response protocols are policies and procedures within schools for responding to crisis situations, which may include responding to a student mental health crisis. Additionally, within schools, educators collaborate with school and community mental health professionals to identify and facilitate access to treatment for students with significant mental health problems and support intensive interventions and needed accommodations.

Educators are well situated to play a role at all tiers within an MTSS with respect to the activities described above. Educators are not expected to take on formal mental health support roles, such as the role of school mental health professionals, and as such, do not provide mental health diagnostic or treatment services for students. However, improving the mental health literacy of educators equips teachers and school staff with the knowledge and skills to take an active role in promoting positive student mental health, identifying students at-risk for mental illness, responding effectively, and collaborating with others when student mental health challenges arise.

### 1.2. Educator Perspectives on Supporting Student Mental Health

Several studies conducted in Canada [14], the United States [15], Australia [16], and Europe [17] have investigated educators’ perspectives on their role in supporting the mental health needs of students. Across studies, teachers generally agreed that promoting and supporting student mental health fits within their role as educators [14,18,19,20,21,22,23]. However, a small number of studies reported variable findings when educators were asked about their role in providing school-based mental health services. For example, a qualitative study of teachers’ perspectives and practices related to school mental health services in urban American secondary schools found that half of the 46 participants viewed supporting student mental health within their role and half viewed this as beyond their scope of practice [15]. Further, a key finding from a qualitative study conducted with 14 teachers in British secondary schools was that participants perceived a majority of teachers were reluctant to engage in emotional health and wellbeing practices with students, despite all participants acknowledging that supporting student mental health was part of their role as educators [24].

Findings across studies indicate that educators viewed themselves assuming a variety of roles in supporting student mental health. Many educators identified their role as offering classroom-based supports for students who may be struggling with a mental health problem, or as providers of class-wide behavioral interventions [15,17,19,20,21,22,25]. Additionally, teachers viewed themselves as collaborators with school- and community-based mental health professionals when supporting student mental health needs [14,15,18]. Findings also suggest that educators considered building relationships with students to be an integral factor in supporting the mental health needs of students [23,24]. In a study conducted by Ekornes [18], educators primarily viewed themselves in the role of a gatekeeper who helped to identify students who may be struggling with their mental health and assist with linking them to appropriate mental health professionals and resources. Although educators identified various roles in supporting student mental health, they also reported a lack of clarity regarding their precise responsibility [15,16,18,23,26]. This reported role uncertainty was identified as one of the significant challenges, among many, that educators experience when attempting to support student mental health.

While several studies found that educators recognized they have an important role in supporting the mental health needs of students, they also identified significant challenges and barriers to effectively serving in this role. Several structural and logistical barriers were identified related to collaborating with other mental health professionals, including time limitations, communication barriers related to confidentiality, and organizational barriers that create separation between educators and mental health professionals [15,22,23,27,28]. Furthermore, educators recognized that when their own emotional needs were neglected, this created a barrier to addressing students’ emotional needs [24,26,29]. Yet, the most commonly cited challenge among educators was a lack of knowledge about mental health, a lack of relevant learning opportunities during pre-service education, and limited ongoing professional development opportunities [16,17,18,19,21,22,23,25,26].

Given educators’ acknowledged limitations related to supporting student mental health, it should not be surprising that further education was an essential need identified in several studies to increase educator confidence and competence in this area [16,17,18,20,22,26]. In a study conducted by Rothi et al. [22], educators specified that they required additional professional learning opportunities related to recognition of mental health problems, provision of classroom supports, relevant school resources and procedures, and appropriate community referral agencies. Another recent study investigated educators’ perceived needs regarding the content and modality of mental health-related professional development and resources [30]. Participants preferred applied education focused on providing concrete strategies that were adaptable based on the developmental level of the students being supported, such as how to approach and have conversations with students about mental health concerns. In terms of educational approach, participants indicated a preference for simple, quick, flexible professional development that incorporated multiple modalities to accommodate diverse learning styles. Furthermore, participants highlighted that interactive learning, incorporating case examples, opportunities for discussion, and built-in checks for understanding were desirable components.

Beyond educational and skill development needs, in several studies educators also acknowledged that they required additional support and collaboration from community and school-based mental health professionals in order to effectively support student mental health needs [14,18,19,21,28,30]. Moreover, educators identify that collaboration with mental health professionals allows for ongoing learning and increasing their own mental health knowledge and skills through the mental health professionals’ expertise [28]. For instance, as part of a comprehensive mental health service network within a school, school psychologists, school counselors, and/or school social workers may be well situated and have the necessary background and expertise to support educators’ mental health literacy practices through professional development, consultation, and ongoing collaboration.

### 1.3. Educator Mental Health Literacy

The construct of mental health literacy evolved from the domain of health literacy, which arose out of recognition that a significant proportion of the population possesses limited knowledge and skills related to maintaining physical health and identifying and seeking treatment for health problems [31]. Furthermore, limited health-related knowledge and skills are associated with numerous poor health outcomes [32]. Similar gaps in knowledge and skills were found to exist in the mental health care field as well, which led to the development of mental health literacy as a distinct construct. Mental health literacy has been defined as the knowledge, skills, and attitudes that allow for effective prevention, identification, and treatment of mental health problems [33]. More recently, definitions of mental health literacy have expanded to include the knowledge and skills that contribute to the promotion of positive mental health [31]. Specifically, important areas of mental health knowledge and skill development include the ability to identify the signs and symptoms of mental health problems, including relevant risk and protective factors [34]. Additionally, mental health literacy includes knowledge about effective treatments and supports, and the skills that allow one to access or help others access mental health supports [31,34]. Furthermore, mental health literacy seeks to instill helpful and accurate beliefs and attitudes about mental health and mental illness to address the widespread misunderstandings held by many members of the general public that contribute to the problem of mental health stigma [31,34]. Finally, mental health literacy includes the skills to maintain positive mental health and wellbeing [34].

When considering the application of mental health literacy within schools, mental health literacy can be considered a foundational component for comprehensive school mental health services [35]. A group of researchers and practitioners with expertise in school-based mental health called the School Based Mental Health and Substance Use Consortium developed a useful working definition, describing school mental health literacy as, 

*the knowledge, skills and beliefs that help school personnel to: create conditions for effective school mental health service delivery; reduce stigma; promote positive mental health in the classroom; identify risk factors and signs of mental health and substance use problems; prevent mental health and substance use problems; help students along the pathway to care*.[35] (p. 4)

As such, educators require skills and strategies for promoting mental health in the classroom, providing social emotional learning instruction, identifying signs of distress in students, and providing accommodations, support, and links to professional services [36]. This comprehensive understanding of school mental health literacy further assumes that developing mental health literacy is a shared responsibility for educators, students, families, school mental health professionals and communities [35].

### 1.4. Mental Health Literacy Professional Development for Educators

It is promising that researchers have identified and begun to take notice of the need for increased educator knowledge related to supporting student mental health. Fortunately, this increased recognition has led to the development, implementation, and evaluation of mental health literacy educational programs to address this gap in educator professional learning. Generally, mental health literacy professional development programs are based on classical theories of behavior change, which posit that increases in knowledge, attitudes, and confidence will result in changes in actions [37]. While multiple mental health literacy programs have been developed in various formats and for use with different populations, the objectives generally remain consistent: (a) through psychoeducation, increase knowledge of how to achieve and maintain positive mental health, and knowledge of common mental health problems; (b) decrease stigma and improve attitudes toward mental illness by illuminating misconceptions related to mental health; (c) increase effective helping behaviors through skill development in identifying and responding to individuals experiencing mental health problems, and supporting access to appropriate services and resources [38,39].

### 1.5. Development of Educator Mental Health Literacy Professional Development Curriculum in the United States

A recent environmental scan of available mental health literacy training for educators revealed a dearth of free and comprehensive mental health literacy training [40]. Findings revealed that although some training was available, it was often costly or did not address all components of mental health literacy (mental health knowledge, mental health promotion, providing support, and reducing stigma). In response, SAMHSA requested that the MHTTC Network develop a comprehensive mental health literacy training package for educators that would be offered at no cost to learners. The request recommended that it be informed by the scientific literature and co-developed with educators to ensure content relevance and acceptability.

## 2. Materials and Methods

### 2.1. Partners

Development of the educator mental health literacy training package was conducted through a partnership between the SAMHSA-funded MHTTC Network (www.mhttc.org (accessed on 15 November 2022)) and the NCSMH (www.schoolmentalhealth.org (accessed on 15 November 2022)), funded by the Health Resources and Services Administration (HRSA). The MHTTC Network is comprised of 10 Regional Centers, a National American Indian and Alaska Native Center, a National Hispanic and Latino Center, and a Network Coordinating Office. It provides workforce development, training, and technical assistance to support the dissemination and implementation of evidence-based practices in the mental health field. In addition, the MHTTC Network has supplemental funding for a school mental health initiative to accelerate the implementation of effective mental health services in schools. The NCSMH’s mission includes strengthening school mental health policies and programs to improve learning and promote success for students. Responding to the need to equip educators with the skills necessary to support student mental health in schools, the MHTTC Network and the NCSMH collaborated in developing a framework and curriculum for a free mental health literacy training package for educators.

### 2.2. Curriculum Development Process

The curriculum development planning process depicted in Figure 1 began with an environmental scan to identify available training programs focused on mental health literacy for educators. The environmental scan process involved conducting literature reviews in the field of school mental health, targeted internet searches, and surveying networks of educators and mental health professionals connected to the MHTTC Network and NCSMH. The results of this environmental scan highlighted a lack of free, comprehensive mental health training programs for educators, as most programs focused on specific topics or had an associated cost [37]. 

### 2.3. NCSMH and MHTTC Network Leadership Brainstorming Session

Next, a two-day brainstorming session was conducted with school mental health leaders from the MHTTC Network and NCSMH. The goal of this session was to review the results of an environmental scan to discern the gaps and needs of educators and develop a preliminary curriculum framework to address this need. Based on the discussions conducted during this brainstorming session, a preliminary conceptual model for a comprehensive educator mental health literacy curriculum was developed, as depicted in Figure 2.

### 2.4. Educator Focus Groups

After conclusion of the initial brainstorming session, three focus groups (*N* = 15) were conducted with educators and school mental health staff from across the United States. Recruitment criteria included current K-12 educators with an interest in the development of a national mental health literacy curriculum and no more than two educators from each MHTTC Region. Participants were recruited by MHTTC School Mental Health Leads from schools throughout the country and included educators and school mental health staff from various roles, grade levels, and communities as illustrated in Table 1. Focus groups were conducted through an online meeting platform and participants were provided a stipend of $100 to compensate their time. Prior to meeting, participants were provided with a brief outline of the course topics and three questions that would be addressed during the focus groups. Focus group questions asked for the most pressing mental health training needs for educators, suggestions for improving or changing the proposed training content, and recommendations for the proposed training format.

A process of Nominal Group Decision Making (NGDM) was used throughout the focus groups [41]. NGDM is a method for group brainstorming that encourages participation from everyone and includes the following steps: (1) introduction and explanation, (2) silent idea generation, (3) sharing ideas, and (4) prioritizing. Though NGDM is commonly used in face-to-face brainstorming sessions, the process was adjusted slightly for an online meeting. After presenting a question to the group, participants were given a moment to independently consider their responses. The moderator, a member of the NCSMH team, then called on each participant in a random order to share their ideas. Each response was typed into a shared document visible by all participants until all ideas were included. Participants were then asked to prioritize their top three responses, which were privately shared with the moderator through the online chat function. After both sessions were completed, participants were asked to complete an online form which allowed them to rank the combined top six responses for each question.

### 2.5. Key Informant Interviews

In addition to the educator focus groups, one-on-one interviews with leaders in the field of school mental health were also conducted. Similar to the educator focus groups, participants were nominated by the MHTTC School Mental Health Leads from regions across the country in an effort to gather a diverse sample. Nine key informants with rich backgrounds in education and/or mental health and various perspectives were identified, including school principals, district leaders, and community mental health partners.

After agreeing to participate in a one-on-one virtual interview, key informants were sent an outline of the proposed training and topics. On the day of the interview, each key informant was asked a series of questions by an NCSMH staff member. In addition to the three questions covered during the focus group interviews, key informants were also asked to identify possible barriers to educators accessing and engaging in the course materials, as well as identifying key resources or tools that should be incorporated into the training. 

Interviews lasted up to one hour and were recorded for notetaking purposes. All responses from interviewees were documented via comprehensive notetaking. Two team members then reviewed the recordings and notes to identify core themes across the interviews (i.e., responses that were mentioned by more than one interviewee). 

## 3. Results

### 3.1. Educator Focus Groups

The top three ranked responses pertaining to each focus group question are listed in Table 2. Participants indicated that the most pressing educator training needs were: (1) recognizing the warning signs of mental illness, (2) strategies for creating classroom environments that promote positive mental health for all students, and (3) classroom strategies for supporting students with trauma backgrounds.

Participants also provided the following suggestions for improving the quality of the training: (1) add content focused on self-regulation, emotion literacy, and empathy development skills, (2) increase the time of the training program to cover content in sufficient depth, and (3) incorporate cultural responsiveness considerations throughout the curriculum.

Finally, participants highly ranked the following suggestions related to the proposed content format: (1) offer a small group or cohort model for the training, (2) include opportunities for questions and answers, and (3) offer guidance for how schools can implement the curriculum during professional development days.

### 3.2. Key Informant Interviews

Responses from key informant interviews are listed in Table 3. Overlapping responses are included as one response and all are abbreviated for conciseness. 

### 3.3. Course Structure and Content

Based on findings from the NCSMH and MHTTC Network brainstorm session, focus groups, and key informant interviews, along with literature on the components of comprehensive mental health literacy, the Classroom Well-Being Instruction and Strategies for Educators (WISE) curriculum was developed. In order to reflect the desire to cover both mental health promotion for all students and support for students in distress, content was divided into two domains focused on each, respectively. Three modules were created for each domain, reflecting the core components of mental health literacy and covering topics indicated by educators through the focus group and key informant interviews. Modules were developed to meet the training length and format desired by informants (i.e., a balance between short and sufficient), include interactive components and videos, and offer supplemental materials to promote deeper learning and facilitate course implementation. Table 4 includes the specific learning objectives for each module.

Following development of the curriculum, the content was translated into a three-part training package, including an online course, video library, and resource collection, all housed on a website, www.classroomwis.org (accessed on 15 November 2022). An instructional design company was engaged to translate the curriculum into a 5 h interactive self-paced online course that includes narration and interactive exercises (e.g., scenario cards, myths versus facts and self-assessment activities). The MHTTC Network and NCSMH worked with a film production company to script, film, and produce a series of 40 short (30–120 s) videos depicting a diverse group of real students and teachers demonstrating strategies and sharing skills to help all students feel supported in the classroom. Finally, a resource collection of over 80 resources, including activities, handouts, and tip sheets was created to allow educators and school staff to bring the content fully into the classroom. An updated theoretical framework for Classroom WISE components and intended outcomes was developed (see Figure 3). 

## 4. Discussion

An MTSS framework to promote student mental health and to address student mental health challenges must include mental health literacy for educators [1,13,18,36]. Educators endorse student mental health as a part of their role, but typically report feeling ill-equipped to offer adequate support as well as experiencing confusion about their role and responsibilities [23,25,30]. Unfortunately, available mental health literacy programs in the United States are limited and are often costly or lacking in a comprehensive approach that includes mental health knowledge, mental health promotion, providing support, and reducing stigma [40]. In response to a request from SAMHSA, the MHTTC Network and the NCSMH co-developed a mental health training package for school staff. 

Scientific literature on mental health literacy in schools, a brainstorming session by NCSMH and MHTTC leadership, focus groups with educators, and key informant interviews informed the structure and content of the resulting course, titled Classroom Well-Being Information and Strategies for Educators (WISE). Course modules reflect the interest in covering general mental health promotion content, including how to create welcoming and supportive classrooms, how to infuse mental health literacy into the curriculum, and how to foster social emotional competencies of students and staff. In addition, modules include content specific to identifying and supporting students exposed to adversity or experiencing psychological distress. 

In response to the desire for practical strategies and interactive formatting, Classroom WISE includes opportunities for learner reflection and response and dynamic videos of educators, students, and experts, some offering demonstrations of specific skills. The development process highlighted an interest in offering a user-friendly instructional platform and supplemental tools for learning and implementation. Therefore, the course was placed on an accessible e-learning platform and includes several extension activities and related resources to deepen learning and facilitate implementation. 

### Limitations

Although Classroom WISE was developed with consideration to the content areas and characteristics that educators and mental health leaders throughout the United States identified as the most imperative to include in a mental health literacy training for educators and school staff, input was compiled from a limited number of focus groups and key informant interviews. Moreover, despite an attempt to obtain input from a demographically diverse group of educators and mental health leaders with varying years of experience and ethnic backgrounds, the participants interviewed were predominantly White women. Additionally, the development of the Classroom WISE online course was constructed based on the input of educators currently working in the United States education system, which does not offer much time or support for mental health literacy training of educators. Thus, the resulting course is only 5 h in length. Furthermore, while Classroom WISE is available free of cost online and on a user-friendly instructional platform for the ease of usability and accessibility to all, and with supplemental online tools for learning and implementation, it does not include live training, which had emerged as a preference during interviews with educators and mental health leaders. Lastly, although the 5-h, self-paced online training may prove advantageous given its feasibility for implementation in the United States learning context, it likely will not sufficiently address all of the unmet mental health training needs of educators. Ideally, educators would receive mental health training throughout their pre-service education and would be offered consistent opportunities for learning and demonstrating mental health literacy skills throughout their career and as part of professional development opportunities.

## 5. Conclusions

Given the deepening mental health crisis among youth, evident before the COVID-19 pandemic, but now worsening, it is critical that educators be provided effective education and training on mental health literacy, so that they can play their essential role in supporting students [1,5,6]. Classroom WISE was developed to equip educators with strategies that can be used to promote student wellbeing and support students experiencing adversity, distress, and mental health challenges in the classroom. 

With the surprising dearth of comprehensive, low- or no-cost options, Classroom WISE offers a systematic and accessible mechanism for providing basic mental health literacy training for educators and school personnel. Classroom WISE was designed to eliminate a number of barriers to provision of educator mental health training that schools face, including cost (Classroom WISE is free); pace (a self-paced course means more accessible learning and retention for a diverse audience); duration (this course is efficiently packed with essential information, averaging 5 h per trainee); and accessibility (available 24/7). In addition, Classroom WISE can be used in a learning community or group format, in person or virtually, by having a group lead walk through each module, with time to discuss practical application and strategies. It can also be used in pre-service education for educators and other school professionals. 

Throughout the process of developing Classroom WISE, educators, school administrators, and students discussed the importance of having culturally inclusive strategies included in mental health literacy training for educators. This reflects the increased attention in the United States to issues of health equity, the effects of trauma and racialized violence, and social determinants of health on children and young adults, as well as in the education system [42,43,44,45]. Although some of this content is included in Classroom WISE, to provide a focused training on the topic the Central East MHTTC Regional Center, in partnership with NCSMH, K-12 educators, and school staff, developed a companion to Classroom WISE, entitled Cultural Inclusiveness and Equity (CIE) WISE. In CIE WISE, educators and school personnel learn how inequities in education impact student mental health and how implicit bias influences our perceptions and responses. Then, building on that foundation, culturally inclusive classroom strategies to support student mental health are presented. Like Classroom WISE, CIE WISE includes a self-paced online course (2 h), video library, and resource collection.

Finally, research on Classroom WISE is ongoing. Participants’ satisfaction with the course, videos, and resources, as well as changes in knowledge, attitudes, and impact on strategies used in the classroom are being evaluated. Additionally, dissemination and implementation science is increasingly being used in school settings to understand how to overcome challenges to uptake of effective practices [46], including those just discussed. Through research in education and health care settings, it’s clear that one course or workshop does not necessarily lead to change in practice or use of a new program [47,48]; therefore, several studies are underway to identify best practices for implementation of Classroom WISE that would assist technical assistance purveyors, including the MHTTC Network, to help school districts and schools adopt this training package and better serve children and adolescents in the location where the majority of mental health services are provided to them, schools. 

## Figures and Tables

**Figure 1 ijerph-20-00035-f001:**
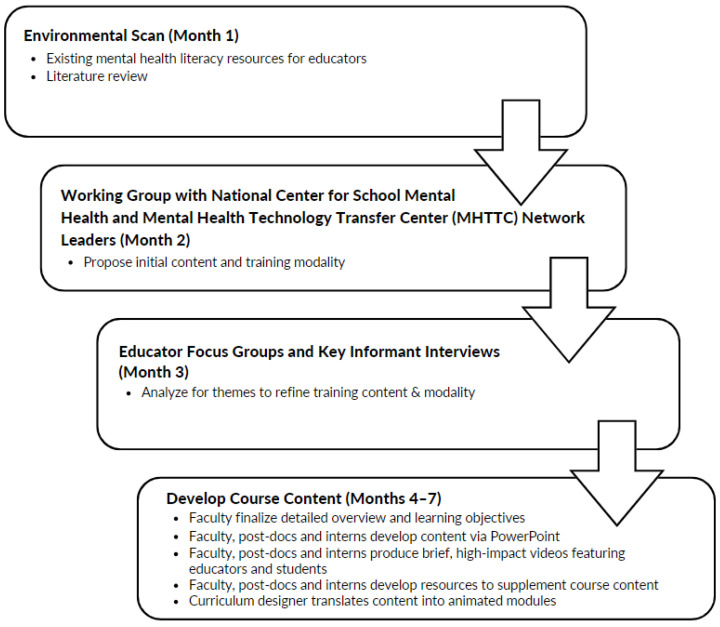
Curriculum development process.

**Figure 2 ijerph-20-00035-f002:**
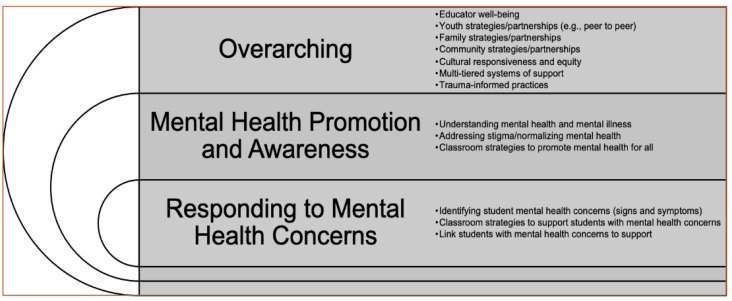
Proposed curriculum content following initial brainstorming session.

**Figure 3 ijerph-20-00035-f003:**
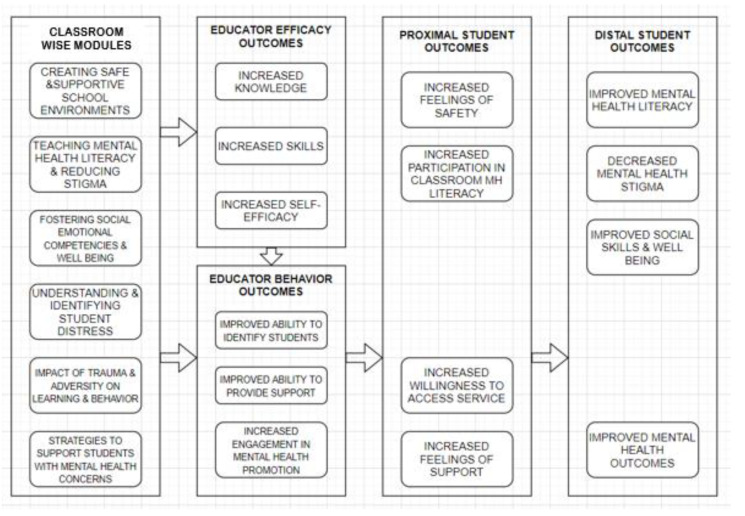
Educator mental health literacy professional development logic model.

**Table 1 ijerph-20-00035-t001:** Focus group participant demographics.

Demographics		N (%)
Race	White or Caucasian	14 (93%)
	Black or African American	1 (7%)
Hispanic/Latino	No	14 (93%)
	Yes	1 (7%)
Gender	Female	13 (87%)
	Male	2 (13%)
Role	Teacher	8 (53%)
	School Administrator	3 (20%)
	Master Teacher/Department Head	2 (13%)
	School Social Worker	1 (7%)
	Other	1 (7%)
Grade Level	Early Childhood	1 (7%)
	Elementary	6 (40%)
	Middle School	2 (13%)
	High School	3 (20%)
	Multiple	3 (20%)
Special Education	Yes	2 (13%)
	No	13 (87%)
School Community	Rural	5 (33%)
	Suburban	3 (20%)
	Urban	4 (27%)
	Multiple	3 (20%)

**Table 2 ijerph-20-00035-t002:** Focus group results.

Rank	Most Highly Endorsed Responses
Most pressing educator training and resource needs related to mental health identification, referral, and supporting student mental health	
1	Recognizing warning signs of mental health issues at various ages (vs. understanding of typical development)
2	How to create a safe and welcoming classroom environment that promotes positive mental health for all kids
3	Interventions that teachers can use in the classroom for trauma
Suggestions to improve the content of the training	
1	Add self-regulation skills/recognizing feelings and emotions/how to develop empathy toward all students (including out of classroom settings)
2	Expand the time of the training to adequately cover content
3	Make sure cultural responsiveness is included and how mental health is viewed in different communities/cultural views
Recommendations for format	
1	Offer a small group or cohort format to compliment the training
2	Include question and answer component
3	Offer guidance to schools on how to best implement the training as part of a professional development day

**Table 3 ijerph-20-00035-t003:** Key informant interview responses.

Question Response
Most pressing educator training and resource needs related to promoting student mental health and well-being
Strategies
Stigma
Common mental health language
Positive mental health (by developmental stage)
Positive, healthy classroom climate
Wellness strategies for students and teachers (including compassion fatigue/secondary trauma)
Most pressing educator training and resource needs related to identification, support, and referral for student mental health challenges
Referral process/who to go to, including the educator’s role in the system
Identifying risk factors without labeling and diagnosing
Screening
Making the training more relevant and helpful for educators
Practical solutions
Concrete examples
Direct connections between mental health and academic success
Appropriate length and format for an online training for educators on student mental health
Short—30 min to 1 h maximum in a single module
Importance of personalization/ability to choose modules based on need, such as one bigger required module and multiple shorter modules that teachers can choose from
Make it interactive
Priorities for mental health topics to be covered in quick (e.g., 1 or 2 min) videos
Anxiety
Internalizing behaviors
Suicide prevention
Educator self-care
Challenges in educators accessing or completing an online mental health training
Time, including allowing teachers to start/stop modules and return later
Buy-in from administration
Tools and/or resources that should be included
Tools for progress monitoring
Handouts/visual tools with simple strategies
Checklists (e.g., behaviors to look for, when to follow-up and refer)

**Table 4 ijerph-20-00035-t004:** Classroom Well-Being Information and Strategies for Educators (WISE) module titles and objectives.

Module #	Module Title	Objectives
1	Creating safe and supportive classrooms	Describe the 3 components necessary for creating safe and supportive classrooms Support students to feel engaged in the classroom community Support students to feel both physically and emotionally safe in the classroom Design a safe and supportive physical classroom environment
2	Teaching mental health literacy and reducing stigma	Describe complete mental health Integrate mental health literacy into instruction Address mental health stigma in the classroom
3	Fostering social emotional competencies and well-being	Define social emotional learning (SEL) Describe the five SEL competencies Integrate SEL competencies into instruction
4	Understanding and supporting students experiencing adversity	Understand and promote healthy child and adolescent development Recognize signs of student distress and who may need additional mental health supports Link students with potential mental health concerns to support
5	Impact of trauma and adversity on learning and behavior	Define childhood trauma and adverse childhood experiences (ACES) Describe the impact of trauma and ACES on learning and overall functioning Demonstrate trauma-sensitive teaching practices
6	Classroom strategies to support students	Understand factors that contribute to student behaviors Practice co-regulation and self-regulation Identify classroom strategies to support students experiencing distress

## Data Availability

Data from focus groups and interviews are securely stored at the University of Maryland, Baltimore.

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
