# Peer review of "Educator-Informed Development of a Mental Health Literacy Course for School Staff: Classroom Well-Being Information and Strategies for Educators (Classroom WISE)"

_ijerph, 2022, doi:10.3390/ijerph20010035_

Round 1
Reviewer 1 Report
You wrote a powerful and very well-structured article that shows how good research can feed into good intervention and vice versa. Engagement of other educational stakeholders beyond school is another aspect I found particularly relevant. More than emphasizing positive aspects in your article I will focus on aspects that you may want to address.
I suggest using the full names of organizations rather than acronyms such as
MHTTC that may not be clear to potential readers.
The theoretical framework is quite sound. There is a book you may want to report to by Nick Clough and Jane Tarr (2022) Addressing issues of mental health in schools through the arts: Teachers and music therapists working together. London: Routledge.
Your article might benefit from a clarification of what you mean by ‘mental health’ and ‘wellbeing’, two concepts that have been quite controversial and seem to be key in your work and feed into even more complex concepts such as ‘comprehensive mental health literacy’. See, for example WHO’s description.
You made recourse to a rich and coherent methodology that surely allowed the teams to get close to the object of research, from the more global, to discussion and group and individual views. Your explanation of the method is clear and detailed. Can you clarify why you included Table 1. Focus group participant demographics? I did not identify if and how you use these demographics in your analysis. Is that because you want to show the diversity of teachers and staff? If that is the case, please say so. How is that relevant for your research and the arguments of the article?
In table 4, I would substitute ‘help’ for ‘support’ to move the action and competence towards subjects/students
In my opinion, the work might be enriched throughout if you positioned yourselves in a clearer way. The description is extremely rich but positioning and questioning might support readers in engaging with what your saying. The introduction of introductory expressions such as ‘surprisingly’ or ‘allowing for deeper reflection on’ might do this job. It may seem a question of style but such approach would emphasize the analytical dimension of your article.
Well done.
Author Response
Thank you for the valuable feedback on the manuscript. Please see our responses to your comments below.
Reviewer 1 Responses
You wrote a powerful and very well-structured article that shows how good research can feed into good intervention and vice versa. Engagement of other educational stakeholders beyond school is another aspect I found particularly relevant. More than emphasizing positive aspects in your article I will focus on aspects that you may want to address.
I suggest using the full names of organizations rather than acronyms such as MHTTC that may not be clear to potential readers.
- Whenever an acronym is used, the term is written out in full when it first appears in the article, with the acronym provided in brackets following introduction on the term. For instance, in Paragraph One of the Introduction, the full terms for Mental Health Technology Transfer Center (MHTTC) and Substance Use and Mental Health Services Administration (SAMHSA) are included.
The theoretical framework is quite sound. There is a book you may want to report to by Nick Clough and Jane Tarr (2022) Addressing issues of mental health in schools through the arts: Teachers and music therapists working together. London: Routledge.
- Thanks for this resource. We will look into this resource for future publications.
Your article might benefit from a clarification of what you mean by ‘mental health’ and ‘wellbeing’, two concepts that have been quite controversial and seem to be key in your work and feed into even more complex concepts such as ‘comprehensive mental health literacy’. See, for example WHO’s description.
- To address this comment, we added the definition and sources below and made adjustments to the reference section: Mental health is often defined and understood in terms of the presence or absence of mental illness. However, increasingly, definitions of mental health incorporate discussion of both the presence of positive indicators of wellbeing (e.g. life satisfaction, self-acceptance, positive emotions), as well as the absence of disorder.
You made recourse to a rich and coherent methodology that surely allowed the teams to get close to the object of research, from the more global, to discussion and group and individual views. Your explanation of the method is clear and detailed. Can you clarify why you included Table 1. Focus group participant demographics? I did not identify if and how you use these demographics in your analysis. Is that because you want to show the diversity of teachers and staff? If that is the case, please say so. How is that relevant for your research and the arguments of the article?
- We added clarification in the manuscript that Table 1 is included to illustrate the variety of participant roles, grade levels, and communities. In the limitations section, we also addressed the limitations of the sample’s demographic diversity illustrated in Table 1.
In table 4, I would substitute ‘help’ for ‘support’ to move the action and competence towards subjects/students
- Done
In my opinion, the work might be enriched throughout if you positioned yourselves in a clearer way. The description is extremely rich but positioning and questioning might support readers in engaging with what your saying. The introduction of introductory expressions such as ‘surprisingly’ or ‘allowing for deeper reflection on’ might do this job. It may seem a question of style but such approach would emphasize the analytical dimension of your article.
- Thank you for this suggestion. We added a few introductory expressions (e.g., “Unfortunately…”; “surprising dearth…”) throughout the manuscript to better engage readers and emphasize the analytic dimension of the article.

Reviewer 2 Report
The paper has scientific content which educators and those who work with kids can use it for promotion of children’s’ mental health. However, I have some comments that may help advancing the paper.
Abstract:
The main items in the curriculum or training packages should be addressed.
Introduction:
In a sentence, it was mentioned “targeting prevention and promotion of mental health…”. Prevention of what? Prevention and promotion of MH?
What is trauma-informed behavioral supports and how can the educators help?
What is crisis response protocols and can educators carry out the protocol or it should be carried out by professionals such as psychologists or MH professionals?
Method:
In participants’ information, the age and years of experience are not reported? Did they were controlled?
More explanation about choosing and entering (the criteria’s) the participants in the study is needed.
More explanation about the informant interviews is needed (what questions were asked or what item were discussed and how the key items were extracted).
Who were in the focus group and how was the process of getting to the key points?
It was mentioned that three modules were created for each domain (address the domains), but in table 4 there are 5 modules; explain it.
Discussion:
More elaboration and discussion about the modules and training package (the activities and the form of their presentation) are needed.
Author Response
Thank you for the valuable feedback on the manuscript. Please see our responses to your comments below.
Reviewer 2
The paper has scientific content which educators and those who work with kids can use it for promotion of children’s’ mental health. However, I have some comments that may help advancing the paper.
Abstract:
The main items in the curriculum or training packages should be addressed.
- The three components of the curriculum (online course, video library, and resource collection) are described in the abstract. This sentence has been added to the abstract: The Classroom WISE curriculum focuses on promoting positive mental health in the classroom, as well as strategies for recognizing and responding to student experiencing mental health related distress.
Introduction:
In a sentence, it was mentioned “targeting prevention and promotion of mental health…”. Prevention of what? Prevention and promotion of MH?
- The sentence has been updated to: As such, targeting prevention of mental health problems and promotion of mental health and wellbeing at the universal level…
What is trauma-informed behavioral supports and how can the educators help?
- We added a definition of trauma-informed behavioral supports. We also added the following sentence as an example of how educators can offer these supports in the classroom: For instance, one example of a trauma-informed behavioural support is the implementation of consistent and predictable classroom routines.
What is crisis response protocols and can educators carry out the protocol or it should be carried out by professionals such as psychologists or MH professionals?
- An additional sentence has been added: Crisis response protocols are policies and procedures within schools for responding to crisis situations, which may include responding to a student mental health crisis.
Method:
In participants’ information, the age and years of experience are not reported? Did they were controlled?
- Age and years experience were not reported because this data was not collected, therefore it was not controlled in the design. More explanation about choosing and entering (the criteria’s) the participants in the study is needed.
- We added a sentence to elaborate on the recruitment criteria of focus group participants: Recruitment criteria included current K-12 educators with an interest in the development of a national mental health literacy curriculum and no more than two educators from each MHTTC Region.
More explanation about the informant interviews is needed (what questions were asked or what item were discussed and how the key items were extracted).
- We provided clarity about the questions that were asked in the key informant interviews: In addition to three questions covered during the focus group interviews, key informants were also asked to identify possible barriers to educators accessing and engaging in the course materials, as well as identifying key resources or tools that should be incorporated into the training.
- We also provided more clarity around the identification of themes (defined as when more than one interviewee identified a topic) by the two independent reviewers: Interviews lasted up to one hour and were recorded for notetaking purposes. All re-sponses from interviewees were documented via comprehensive notetaking. Two team members then reviewed the recordings and notes to identify core themes across the inter-views (i.e., responses that were mentioned by more than one interviewee).
Who were in the focus group and how was the process of getting to the key points?
- The participants in the focus groups are described in section 2.4 and Table 1. The focus group interview process is described in Paragraph One on page 8.
“A process of Nominal Group Decision Making (NGDM) was used throughout the focus groups [38]. NGDM is a method for group brainstorming that encourages participation from everyone and includes the following steps: (1) introduction and explanation, (2) silent idea generation, (3) sharing ideas, and (4) prioritizing. Though NGDM is commonly used in face-to-face brainstorming sessions, the process was adjusted slightly for an online meeting. After presenting a question to the group, participants were given a moment to independently consider their responses. The moderator, a member of the NCSMH team, then called on each participant in a random order to share their ideas. Each response was typed into a shared document visible by all participants until all ideas were included. Participants were then asked to prioritize their top three responses, which were privately shared with the moderator through the online chat function. After both sessions were completed, participants were asked to complete an online form which allowed them to rank the combined top six responses for each question.”
It was mentioned that three modules were created for each domain (address the domains), but in table 4 there are 5 modules; explain it.
- Following the initial brainstorming session, 3 modules were proposed (See Figure 2). After sharing the proposed curriculum plan with the focus groups and key informant interview participants and responding to their feedback, the updated curriculum structure was developed. This is described in section “3.3 Course Structure and Content” on Page 10 of the manuscript. Table 4 includes all 6 modules.
“Based on findings from the NCSMH and MHTTC Network brainstorm session, focus groups, and key informant interviews, along with literature on the components of comprehensive mental health literacy, the Classroom Well-Being Instruction and Strategies for Educators (WISE) curriculum was developed. In order to reflect the desire to cover both mental health promotion for all students and support for students in distress, content was divided into two domains focused on each, respectively. Three modules were created for each domain, reflecting the core components of mental health literacy and covering topics indicated by educators through the focus group and key informant interviews. Modules were developed to meet the training length and format desired by informants (i.e., a balance between short and sufficient), include interactive components and videos, and offer supplemental materials to promote deeper learning and facilitate course implementation. Table 4 includes the specific learning objectives for each module.”
Discussion:
More elaboration and discussion about the modules and training package (the activities and the form of their presentation) are needed.
- In the Discussion, we provide detail about the curriculum content and the training package activities:
Course modules reflect the interest in covering general mental health promotion content, including how to create welcoming and supportive classrooms, how to infuse mental health literacy into the curriculum, and how to foster social emotional competencies of students and staff. In addition, modules include content specific to identifying and supporting students exposed to adversity or experiencing psychological distress.
In response to the desire for practical strategies and interactive formatting, Classroom WISE includes opportunities for learner reflection and response and dynamic videos of educators, students, and experts, some offering demonstrations of specific skills. The development process highlighted an interest in offering a user-friendly instructional platform and supplemental tools for learning and implementation. Therefore, the course was placed on an accessible e-learning platform and includes several extension activities and related resources to deepen learning and facilitate implementation.
